# A Quality-Diversity Controllable GAN for Text Generation

## Abstract

Text generation is a critical and difficult natural language processing task. Maximum likelihood estimate (MLE) based models have been arguably suffered from exposure bias in the inference stage and thus varieties of language generative adversarial networks (GANs) bypassing this problem have emerged. However, recent study has demonstrated that MLE models can constantly outperform GANs models over quality-diversity space under several metrics. In this paper, we propose a quality-diversity controllable language GAN. We illustrate an intimate connection between forward/reverse KL divergences and our principled approach, and demonstrate that our proposed model can achieve quality-diversity control by adjusting a hyper-parameter. Through extensive experiments, we show that the proposed model outperforms MLE models and other GANs variants on synthetic data and is more competitive on real-world tasks such as COCO Image Captions and EMNLP2017 WMT News datasets.

## 1 Introduction

Text generation is one of the most challenging and widely used natural language processing task and it serve as a critical module on numerous practical applications. In recent years, neural autoregressive models have shown superiority on various natural language generation (NLG) tasks (Zhang & Lapata, 2014; Gal & Ghahramani, 2016; Du et al., 2017; Fiorini & Lu, 2018). The most common method of training these models is the maximum likelihood estimation (MLE) which maximize the log predictive likelihood of each true token in the training sequence given the previous observed tokens. However, MLE models have been arguably suffered from *exposure bias* (Bengio et al., 2015; Ranzato et al., 2016) which means a discrepancy between training and inference stages due to the fact that at inference time, the model predicts next token based on the sequence generated by itself which may not occur in training set. In order to overcome this problem, scheduled sampling (Bengio et al., 2015) and professor forcing (Lamb et al., 2016) are proposed, but scheduled sampling is proven to be fundamentally inconsistent (Huszár, 2015) and professor forcing use an adversarial method similar to GANs (Goodfellow et al., 2014).

Varieties of generative adversarial networks (GANs) (Goodfellow et al., 2014) models for text generation have emerged as alternatives to MLE models with the hope of not suffering from exposure bias. However, due to the discrete nature, GANs have difficulty in text generation since the gradients from discriminator can not be passed to generator explicitly and thus most these GANs based models (Yu et al., 2017; Guo et al., 2018; Lin et al., 2017; Fedus et al., 2018; Xu et al., 2018) treat text generation as a sequential decision making process (Bachman & Precup, 2015) and utilize policy gradient (Williams, 1992) to overcome this difficulty. Despite the impressive achievements, they inevitably have a problem of high variance due to the use of reinforcement learning (RL) strategies.

Caccia et al. (2018) have made several surprising observations and argued that, through simple temperature sweeping at inference stage, MLE models can constantly outperform these RL-based GANs models by a large margin over quality-diversity space under several metrics. They attribute this to the fact that the negative impact coming with RL is severer than so-called exposure bias caused by MLE. A RL-free approach to train GANs for text generation without resorting to an explicit neural network as the discriminator, named No Neural Network as the Discriminator GAN (N3DGAN), is proposed by (Li et al., 2019) and shows state-of-the-art results on both synthetic and real datasets under quality-only metrics.

We propose a quality-diversity controllable GAN through a principle approach which is a generalization of N3DGAN (Li et al., 2019). This generalization has intimate connections with direct optimization of both forward and inverse KullbackLeibler (KL) divergences with respect to empirical distribution and model distribution. As shown in (Huszár, 2015), minimizing forward KL divergence often leads to models that overgeneralize, and sometimes produce samples that are very unlikely under true distribution and thus it is good at diversity. Minimizing the reverse KL divergence instead will try to avoid any behaviour that is unlikely under true distribution at the cost of ignoring modes of true distribution completely, thus it is good at quality. Through this principled approach, we demonstrate that our proposed model can control the quality-diversity trade-off, that is, the degree to which the model depends on forward and reverse KL divergences respectively through a single hyper-parameter.

The key contributions of our work are three-folds: (1) We propose a simple but effective GANs model for text generation which can control the quality-diversity trade-off through a single hyper-parameter at training stage. (2) Different from most existing GANs variants, our proposed model, as a generalization of N3DGAN, does not need to rely on MLE pre-training and is RL-free. (3) Through extensive experiments, we show that our proposed model outperform MLE based model and other GANs variants over quality-diversity space in the synthetic-data tasks and are more competitive to MLE model than other GANs variants in the real-world tasks.

## 2    RELATED WORK

Huszár (2015) analyzes the difference between optimizing forward and reverse KL divergences, and suggests to use a generalized Jensen-Shannon divergence to replace MLE, but he doesn't realize its relation to the mini-max game's value function as well as tractability of directly optimizing the generalized Jensen-Shannon divergence on training data, and moreover he doesn't conduct any experiments.

Caccia et al. (2018) try to navigate the quality-diversity space through temperature sweep, beam search and generator rejection sampling (Ott et al., 2018) methods, unfortunately all these methods can only navigate the quality-diversity space at inference stage.

Li et al. (2019) propose N3DGAN which is RL-free and doesn't need additional maximum likelihood pre-training, but they only report results on quality metrics but not on diversity.

Lu et al. (2019) propose a cooperative training (CoT) approach for text generation which dont have to rely on RL and MLE for pre-training, instead they optimize an approximate Jensen-Shannon divergence. It is done by introducing a mediator and dropping the term in the gradient of generator which behaves like RL. In the synthetic-data experiments, they perform similarly but slightly worse than MLE model (Caccia et al., 2018).

de Masson d'Autume et al. (2019) combine existing techniques such as large batch size, dense rewards and discriminator regularization to train language GANs from scratch without the need for MLE pre-training. But they still have to use RL which is essentially the same as most language GANs (Yu et al., 2017; Guo et al., 2018; Lin et al., 2017; Fedus et al., 2018; Xu et al., 2018), and they don't conduct any experiment on synthetic data which has oracle metrics to validate its effectiveness.

## 3    A QUALITY-DIVERSITY CONTROLLABLE GAN

In the well-known adversarial modeling framework Goodfellow et al. (2014), there are two players, one is called generator represented by a differentiable function $G$ and another one is called discriminator represented by a differentiable function $D$. They approximate the data distribution $p_{data}$ through an alternative mini-max game between these two components. The discriminator $D$ is trained to discriminate the samples drawn from the generator distribution $p_G$ from the real data, while the generator is trained to fool the discriminator so that it cannot distinguish between the samples from $p_G$ and $p_{data}$. Goodfellow et al. (2014) proved that if the generator $G$ has enough capacity to cover data distribution $p_{data}$, then optimizing the mini-max game is equivalent to directly optimizing the Jensen-Shannon divergence. Unfortunately Goodfellow et al. (2014) didn't distinguish the real data as either population (*true but unknown*) distribution or empirical distribution.

Li et al. (2019) notice the distinction and they explicitly use the empirical distribution in their approach. They argue that when an alternative mini-max optimization procedure is performed for the value function, a closed form solution for the discriminator exists in the maximization step, thus an explicit neural network model for the discriminator is not needed. Moreover optimizing the mini-max game is equivalent to directly optimizing the Jensen-Shannon divergence (JSD) between the generator's distribution and the empirical distribution over the training data, where the corresponding KL divergence related to the generator's distribution and the empirical distribution over the training data is reduced to the sum over the training data. This means that sampling from the generator over the entire space as well as the policy-gradient algorithm to compute the gradient of the generator can be eliminated. Optimizing the JSD becomes computationally tractable to train the generator for text generation.

We now propose a quality-diversity controllable GAN which is a generalization of N3DGAN (Li et al., 2019) and illustrate its connections to the forward KL divergence and the reverse KL divergence as stated below.

Since the samples in the training data are limited, we use $\pi \in [0, 1]$ to denote the ratio of labeled samples discriminator receives from the generator and the training data, and let the discriminator $D$ and the generator $G$ play the following two-player minimax game with a *generalized* distinguishability game value function $V_\pi(G, D)$:

$$\min_G \max_D V_\pi(G, D) = \pi \cdot \mathbb{E}_{x \sim \tilde{p}_{\text{data}}(x)}[\log D(x)] + (1 - \pi) \cdot \mathbb{E}_{x \sim p_G(x)}[\log(1 - D(x))] \quad (1)$$

where $\tilde{p}_{\text{data}}(x)$ denotes the empirical distribution over training set $\mathcal{C} = \{x_1, \cdots, x_N\}$ with $N$ samples, and

$$\tilde{p}_{\text{data}}(x) = \begin{cases} \frac{1}{N} & \text{if } x \in \mathcal{C} \\ 0 & \text{otherwise} \end{cases}$$

We will then show that optimize (1) is equivalent to optimize generalized Jensen-Shannon divergence (Huszár, 2015) between $\tilde{p}_{data}$ and $p_G$, which is formulated as follows

$$JSD_\pi(\tilde{p}_{\text{data}}(x) \parallel p_G(x))$$
$$= \pi D_{KL}(\tilde{p}_{\text{data}}(x) \parallel \pi \tilde{p}_{\text{data}}(x) + (1 - \pi)p_G(x)) + (1 - \pi)D_{KL}(p_G(x) \parallel \pi \tilde{p}_{\text{data}}(x) + (1 - \pi)p_G(x))$$

and we can control the quality-diversity trade-off through a single parameter $\pi$.

**Proposition 1.** When optimizing the value function $V_\pi(G, D)$, for any given generator $G$, the optimal discriminator $D_G^*(x)$ is

$$D_G^*(x) = \begin{cases} \frac{\pi \tilde{p}_{\text{data}}(x)}{\pi \tilde{p}_{\text{data}}(x) + (1 - \pi)p_G(x)} & \text{if } x \in \mathcal{C} \\ 0 & \text{otherwise} \end{cases}$$

and the value function at the optimal discriminator $D_G^*(x)$ becomes

$$V_\pi(G, D_G^*(x))$$
$$\propto JSD_\pi(\tilde{p}_{\text{data}}(x) \parallel p_G(x))$$
$$\propto \pi \sum_{x \in \mathcal{C}} \tilde{p}_{\text{data}} \log \frac{\tilde{p}_{\text{data}}(x)}{\pi \tilde{p}_{\text{data}}(x) + (1 - \pi)p_G(x)} + (1 - \pi) \sum_{x \in \mathcal{C}} p_G(x) \log \frac{p_G(x)}{\pi \tilde{p}_{\text{data}}(x) + (1 - \pi)p_G(x)}$$
$$+ (1 - \pi)\log(1 - \pi) \sum_{x \in \mathcal{C}} p_G(x) \quad (2)$$

**Proof:** Given any generator $G(x)$, the training criterion for the discriminator $D(x)$ is to maximize the value function $V_\pi(G, D)$ in (1) and could be re-written as.

$$V_\pi(G, D) = \pi \sum_{x \in \mathcal{C}} \tilde{p}_{\text{data}} \log D(x) + (1 - \pi) \sum_{x \in \mathcal{C}} p_G(x) \log(1 - D(x))$$
$$+ (1 - \pi) \int_{x \notin \mathcal{C}} p_G(x) \log(1 - D(x))dx \quad (3)$$

Taking derivative with respect to $D(x)$, we have

$$\frac{\partial V_\pi(G, D)}{\partial D(x)} = \begin{cases} \pi \frac{\tilde{p}_{\text{data}}(x)}{D(x)} - (1 - \pi) \frac{p_G(x)}{(1 - D(x))} & \text{if } x \in \mathcal{C} \\ -(1 - \pi) \frac{p_G(x)}{(1 - D(x))} & \text{otherwise} \end{cases} \tag{4}$$

Thus, the optimal solution for $D(x)$ is

$$D_G^*(x) = \begin{cases} \frac{\pi \hat{p}_{\text{data}}(x)}{\pi \hat{p}_{\text{data}}(x) + (1 - \pi) p_G(x)} & \text{if } x \in \mathcal{C} \\ 0 & \text{otherwise} \end{cases} \tag{5}$$

Plug (5) into (3), we have

$$V(G, D_G^*(x))$$
$$= \pi \sum_{x \in \mathcal{C}} \tilde{p}_{\text{data}} \log \frac{\tilde{p}_{\text{data}}(x)}{\pi \tilde{p}_{\text{data}}(x) + (1 - \pi) p_G(x)} + (1 - \pi) \sum_{x \in \mathcal{C}} p_G(x) \log \frac{p_G(x)}{\pi \tilde{p}_{\text{data}}(x) + (1 - \pi) p_G(x)}$$
$$+ (1 - \pi) \log(1 - \pi) \sum_{x \in \mathcal{C}} p_G(x) + \pi \log \pi \sum_{x \in \mathcal{C}} \tilde{p}_{\text{data}}(x) \tag{6}$$

The generalized JSD can be written as

$$JSD_\pi(\tilde{p}_{\text{data}}(x) \parallel p_G(x))$$
$$= \pi D_{KL}(\tilde{p}_{\text{data}}(x) \parallel \pi \tilde{p}_{\text{data}}(x) + (1 - \pi) p_G(x)) + (1 - \pi) D_{KL}(p_G(x) \parallel \pi \tilde{p}_{\text{data}}(x) + (1 - \pi) p_G(x))$$
$$= \pi \sum_{x \in \mathcal{C}} \tilde{p}_{\text{data}} \log \frac{\tilde{p}_{\text{data}}(x)}{\pi \tilde{p}_{\text{data}}(x) + (1 - \pi) p_G(x)} + (1 - \pi) \sum_{x \in \mathcal{C}} p_G(x) \log \frac{p_G(x)}{\pi \tilde{p}_{\text{data}}(x) + (1 - \pi) p_G(x)}$$
$$+ (1 - \pi) \log(1 - \pi) \sum_{x \in \mathcal{C}} p_G(x) - (1 - \pi) \log(1 - \pi) \tag{7}$$

Because the last terms on the right side of Equation (6) and Equation (7) both are constant, thus equation (2) is established.

Proposition 1 shows that optimizing the mini-max game of equation (1) is equivalent to optimizing equation (2) which is over the training data, thus it is tractable and the model parameter $\theta$ can be updated directly using stochastic gradient descent (SGD) or its variants.

**Proposition 2.** When $\pi$ is converging to 0, optimizing $\frac{V_\pi(G,D)}{\pi}$ is equivalent to optimizing the forward KL divergence $D_{KL}(\tilde{p}_{\text{data}}(x) \parallel p_G(x))$, i.e. optimizing maximum log-likelihood; and when $\pi$ is converging to 1, optimizing $\frac{V_\pi(G,D)}{1-\pi}$ is equivalent to optimizing the reverse KL divergence $D_{KL}(p_G(x) \parallel \tilde{p}_{\text{data}}(x))$.

**Proof:** Consider the Equation (2), when $\pi \to 0$, the second and third terms of $\frac{V_\pi(G,D)}{\pi}$ is equivalent to 0 and thus we have

$$\lim_{\pi \to 0} \frac{V_\pi(G, D)}{\pi} = \lim_{\pi \to 0} \sum_{x \in \mathcal{C}} \tilde{p}_{\text{data}} \log \frac{\tilde{p}_{\text{data}}(x)}{\pi \tilde{p}_{\text{data}}(x) + (1 - \pi) p_G(x)} = D_{KL}(\tilde{p}_{\text{data}}(x) \parallel p_G(x)) \tag{8}$$

The same can be proved that in the limit of $\pi \to 1$, $\frac{V_\pi(G,D)}{1-\pi}$ is equivalent to reverse KL divergence $D_{KL}(p_G(x) \parallel \tilde{p}_{\text{data}}(x))$.

This proposition shows an intimate connection of optimizing the objective $V_\pi(G, D)$ to those of the forward and reserve KL divergences. Thus we can control the quality-diversity trade-off for text generation through a single hyper-parameter $\pi$.

## 4 EXPERIMENTAL STUDY

### 4.1 DATASETS

The experiments consist of two parts: synthetic-data experiments and real-world experiments. We use two datasets, COCO Image Captions dataset and EMNLP2017 WMT News dataset in our real-world experiments since they have become common benchmarks for text generation (Guo et al., 2018; Caccia et al., 2018; Montahaei et al., 2019).

### 4.1.1 SYNTHETIC DATASET

In synthetic-data experiments, in order to test the validity of our proposed model we conduct a simulated test similar to Yu et al. (2017). Following them, we use a random initialized LSTM as the oracle model to generate the real data distribution $P_{oracle}(x_t|x_1, \cdots, x_{t-1})$ to stimulate the real-world structured sequences. We use the oracle model to generate 10,000 sequences of length 20 and 40 respectively as our training set and another two 5,000 sequences are generated as our validation set and test set. Both of them have a vocab size of 5,000.

### 4.1.2 COCO IMAGE CAPTIONS DATASET

For COCO Image Captions dataset (Chen et al., 2015), we use the same pre-processing method as (Guo et al., 2018), but we randomly select 5,000 samples from training set as our validation set. The resulted dataset consist of a training set with 75,000 samples, a validation set with 5,000 samples and a test set with 5,000 samples. The vocab size is 4,838 and the maximum length of captions is 32.

### 4.1.3 EMNLP2017 WMT NEWS DATASET

EMNLP2017 WMT News (Bojar et al., 2017) is a collection of news texts for the machine translations task, among a version of this dataset for English Corpus containing more than 600,000 sentences, we drop the sentences having more than 3 words with less than 400 frequency and keep only the sentences that have between 20 and 40 words. We replace these most infrequent words with token $< unk >$. Next, we randomly sample 200,000 sentences as the training set, 10,000 sentences as the validation set and 10,000 as the test set. The resulted vocab size is 3,980.

## 4.2 EVALUATION METRICS

It is still difficult to accurately evaluate the effects of text generation models using existing metrics since the effects consist of several aspect such as consistency, diversity and fluency. Thus, in order to be consistent, we choose evaluation metrics according to previous work (Semeniuta et al., 2018; Caccia et al., 2018; de Masson d'Autume et al., 2019; Montahaei et al., 2019). We use negative log likelihood for synthetic-data experiments and use n-gram based metrics and Fréchet Distance (FD) for real-data experiments.

### 4.2.1 NEGATIVE LOG LIKELIHOOD

In the work of (Yu et al., 2017), the authors recommend to use the oracle negative log likelihood ($\text{NLL}_{oracle}$) which is an oracle quality metric for synthetic-data experiments. However, only use this metric will be misleading since it is blind to diversity. Thus, similar to Caccia et al. (2018), we additionally use a metric $\text{NLL}_{test}$ to represent diversity that is calculated by taking as input the sentences generated by oracle model to the language model we're interested in.

### 4.2.2 N-GRAM BASED METRICS

We use the local metrics BLEU-5 (Papineni et al., 2002) and Self BLEU-5 (Zhu et al., 2018) to capture quality and diversity respectively. While BLEU based metrics is considered to have limitations that only capture local consistency and fail to capture semantic variations (Semeniuta et al., 2018; Caccia et al., 2018; de Masson d'Autume et al., 2019), they are still one of the most widely used and simple metrics for text generation tasks.

### 4.2.3 FRÉCHET BERT DISTANCE

Fréchet Inception Distance (FID) is originally proposed to computes the distance between two Gaussian distributions of the features of real images and generated images extracted by Inception network (Heusel et al., 2017). We follow Montahaei et al. (2019) to utilize BERT (Devlin et al., 2019), a powerful unsupervised pre-training model for text, to replace the Inception network to capture quality and diversity of generated sentences. This metric is named as Fréchet BERT Distance (FBD).

We do not use MS-Jaccard and Language Model Score (LMS) because MS-Jaccard is also an n-gram based metric like BLEU and it is highly correlated with FBD (Montahaei et al., 2019), and LMS relies on training new models and have an inherent bias favoring MLE language models, since they are trained using the same criteria (de Masson d'Autume et al., 2019).

### 4.3 IMPLEMENTATION DETAILS

We choose our hyper-parameters according to the result on validation set and report all metrics on test set. Our models are implemented with Tensorflow (Abadi et al., 2016) using an NVIDIA V100 GPU. The hidden size, embedding size and batch size we used are 32, 32 and 64 respectively which are kept the same as Texygen framework (Zhu et al., 2018) for fair comparison. We terminate the training of all models based on $NLL_{oracle}$ in synthetic-data experiments and based on their BLEU-5 scores on validation set in real-data experiments.

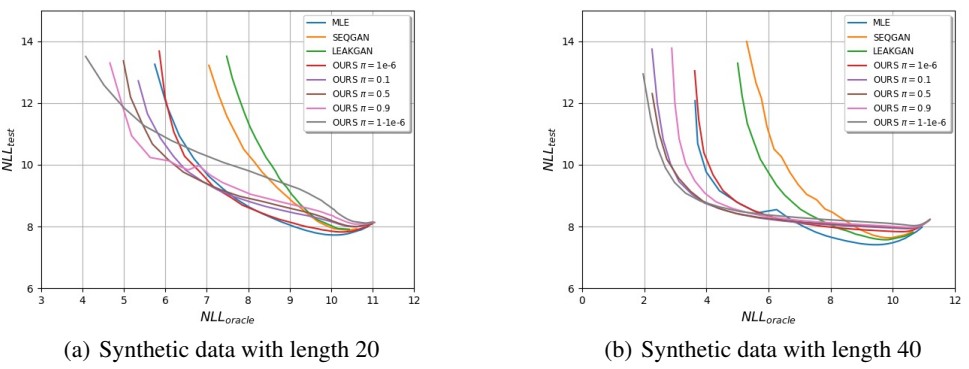

(a) Synthetic data with length 20    (b) Synthetic data with length 40

Figure 1: Temperature sweep curves for the synthetic-data experiments of sentence length 20 (left) and 40 (right) over $NLL_{test}$ vs. $NLL_{oracle}$ space (lower is better for both metrics)

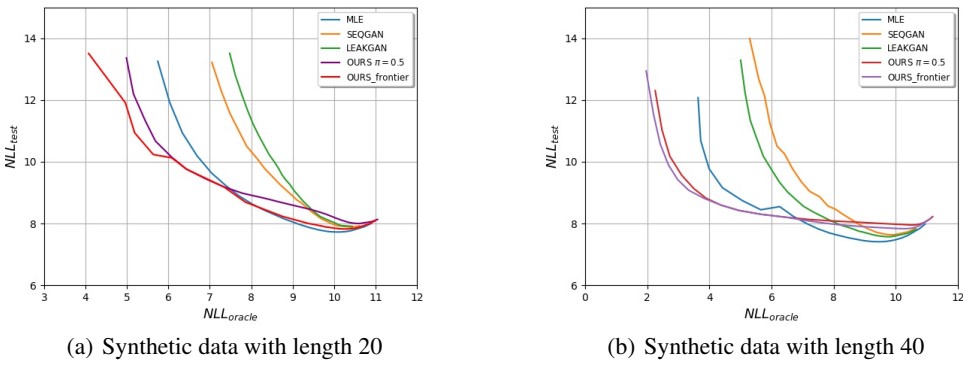

(a) Synthetic data with length 20    (b) Synthetic data with length 40

Figure 2: Frontier curves of our models and three baseline models. Left is better and down is better. The models trained via our proposed adversarial objective perform better compared with MLE model and other GANs variants.

### 4.4 EXPERIMENTAL RESULTS

In this section, we report the experimental results on synthetic-data experiments and real-world experiments. We compare our proposed models with three baseline models: MLE, SeqGAN (Yu et al., 2017), LeakGAN (Guo et al., 2018) in all the experiments. Similar to Caccia et al. (2018),

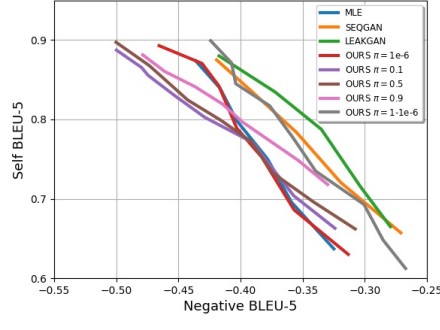

(a) COCO Image Captions dataset

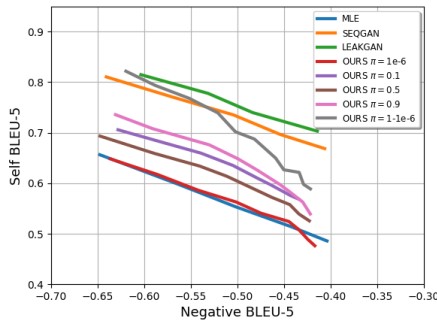

(b) EMNLP2017 WMT News dataset

Figure 3: Negative BLEU-5 plotted against Self BLEU-5 for COCO Image Captions (left) and EMNLP2017 WMT News (right) test set. Lower is better for both metrics.

| Models | Temperature | | | | | |
| | 0.7 | 0.8 | 0.9 | 1.0 | 1.1 | 1.2 |
|---|---|---|---|---|---|---|
| MLE | 4.247 | **4.125** | **4.064** | 4.067 | 4.141 | 4.346 |
| SEQGAN | 4.783 | 4.778 | 4.811 | 4.957 | 5.171 | 5.425 |
| LEAKGAN | 5.585 | 5.580 | 5.692 | 5.197 | 5.441 | 5.702 |
| OURS $\pi \to 0$ | **4.231** | 4.128 | 4.076 | 4.088 | 4.162 | 4.325 |
| OURS $\pi = 0.1$ | 4.398 | 4.267 | 4.113 | 4.065 | 4.031 | **4.129** |
| OURS $\pi = 0.5$ | 4.443 | 4.234 | 4.122 | 4.053 | **4.024** | 4.147 |
| TRAINING SET | - | - | - | 3.581 | - | - |

Table 1: FBD scores on COCO Image Captions test set at different temperatures.

we conduct sweep of temperatures in inference stage to evaluate the trained models over quality-diversity space. We fixed the hyper-parameter $\pi$ and sweep the temperatures so that we can get different temperature curves for our proposed models trained with different $\pi$. Samples from our proposed model and baseline models are shown in Appendix A. According to the generated samples, we can observe that our proposed model outperforms SeqGAN and LeakGAN, and when $\pi \to 1$, it can be seen that the samples are more focused on some of the topics and have good quality, while when $\pi \to 1$, the topics of the samples are more diverse and have better diversity.

### 4.4.1 SYNTHETIC EXPERIMENTS

In the synthetic-data experiments, we use $\text{NLL}_{oracle}$ and $\text{NLL}_{test}$ as our quality and diversity metrics respectively. Both of them are the lower the better. Lowering the temperature make the model tend to produce high quality but similar sentences, while raising the temperature makes the model tend to generate diverse but relatively random sentences. We compute all metrics at different temperatures for each model to draw sweep curves in quality-diversity space.

Figure 1 depicts the temperature sweep curves of sequence length 20 and 40 respectively over $\text{NLL}_{test}$vs.$\text{NLL}_{oracle}$ space. From this figure, we can observe that when $\pi \to 0$, the curve of our proposed model basically overlaps with the curve of the MLE model which shows the correctness of proposition 2. Besides, we can also observe that our models outperform other GANs variants by a large margin and perform much better than MLE on quality metric at the cost of relatively small losses on diversity metric. It is worth noting that our models do not need additional maximum likelihood pre-training and reinforcement learning and thus it is more faster in training than traditional GANs variants. As can be seen from Figure 2, the frontier curve obtained from a family of temperature sweep curves of our models outperforms the curves of MLE, SeqGAN and LeakGAN in most cases.

| Models | Temperature | | | | | |
|--------|------|------|------|------|------|------|
| | 0.7 | 0.8 | 0.9 | 1.0 | 1.1 | 1.2 |
| MLE | 6.263 | 5.524 | **5.067** | 4.816 | 5.204 | **5.726** |
| SEQGAN | 7.423 | 6.674 | 6.321 | 5.932 | 6.268 | 6.814 |
| LEAKGAN | 7.632 | 6.962 | 6.510 | 6.176 | 6.560 | 7.138 |
| OURS $\pi \to 0$ | **6.260** | **5.513** | 5.082 | 4.803 | **5.153** | 5.733 |
| OURS $\pi = 0.1$ | 6.784 | 6.127 | 5.663 | 5.341 | 5.754 | 6.244 |
| OURS $\pi = 0.5$ | 6.660 | 6.004 | 5.542 | 5.241 | 5.640 | 6.155 |
| TRAINING SET | - | - | - | **3.317** | - | - |

Table 2: FBD scores on EMNLP2017 WMT News test set at different temperatures.

### 4.4.2 REAL-DATA EXPERIMENTS

Figure 3 reports negative BLEU-5 versus Self BLEU-5 metrics on COCO Image Captions and EMNLP2017 WMT News test sets for the models trained via our proposed objective, MLE model and other GANs variants. Table 1 and 2 show the FBD scores on these two test sets respectively for the models and training set.

For COCO Image Captions task, as can be seen from Figure 3(a) and Table 1, when hyper-parameter $\pi = 0.1$ and $\pi = 0.5$, our models outperform other GANs variants under different quality-diversity spaces and outperform MLE model in terms of quality metric, which may be because compared with maximum likelihood, our models can properly optimize both forward KL divergence $D_{KL}(\tilde{p}_{\text{data}}(x) \parallel p_G(x))$ and reverse KL divergence $D_{KL}(p_G(x) \parallel \tilde{p}_{\text{data}}(x))$ and compared with other GANs variants, we do not rely on reinforcement learning. When hyper-parameter $\pi \to 0$, the result shown in Figure 3(a) and Table 1 demonstrate that the performance of our model is almost identical to the MLE model which is consistent with proposition 2 and better than other GANs variants .

For EMNLP2017 WMT News task, as shown in Figure 3(b) and Table 2, when $\pi = 0.1$ is and $\pi = 0.5$, although our models does not perform as well as the MLE model, they are significantly better than other GANs variants and more competitive to MLE model.Consistent with the conclusions obtained from the synthetic data experiments and the COCO experiment, the performances of our proposed model and the MLE model are almost equal when $\pi$ tends to zero. Compared with the COCO task, the difference between the FBD scores of the models on the News task and training set is slightly large which may be because EMNLP2017 WMT News is a relatively more difficult task.

## 5 CONCLUSION

This research proposes a quality-diversity controllable GAN which is a generalization of N3DGAN and it can achieve quality-diversity control through a single hyper-parameter $\pi$. When $\pi$ converges to 0, it is equivalent to directly optimizing forward KL divergence; and when $\pi$ converges to 1, it is equivalent to directly optimizing reverse KL divergence. Through experiments in synthetic-data and real-data scenarios, we verify the convergence and the controllability, and we show that our proposed model outperforms MLE model and other GANs variants over quality-diversity space in the synthetic-data experiments, and in the real-data experiments, it is more competitive to MLE model than other GANs variants under several metrics. As can be seen from the experimental results, depending on the task and purpose, such as to get higher quality text or more diverse text, we should choose different hyper-parameter $\pi$ to get better results on what we are interested in.

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

## A   SAMPLES

| Sources | Example |
|---|---|
| OURS $\pi \to 0$ | A person is sitting on top of a wooden table . |
| | A boy with a blue kite on the beach holding his child . |
| | A little girl holding a kite in a grassy field . |
| | A young boy flying a kite in front of a colorful kite . |
| | A man wearing a girl holding plant racer in the air . |
| | A girl with a large body of people flying kites in the sky . |
| | Two people standing in a living room with food in the open . |
| | A tall dog laying on top of a tennis court . |
| | A couple of people standing next to a field playing objects wade . |
| | A man and boy jumping in front of a large chair . |
| OURS $\pi \to 1$ | A couple of people standing on a beach . |
| | A young child flying a kite sitting on a beach . |
| | A girl is standing on a lush green field . |
| | A group of people on top of a sandy beach . |
| | A crowd of people flying a kite in a snow slope . |
| | A person is flying a kite laying on a sandy beach . |
| | A person is holding a kite in a garden . |
| | Two women sitting on top of a sandy beach . |
| | A group of people in a field . |
| | A man standing on the beach next to the water . |
| MLE | A woman in a wide picture holding a kite with a kite . |
| | A woman in a living room next to a large window . |
| | A man using a cell phone holding a kite . |
| | A tennis player is flying a kite in a field . |
| | A man and a girl take a picture to fly a kite . |
| | A man with a child in a park while a woman watches . |
| | Three people and children sitting on chairs , talking on a cellphone . |
| | A crowd of people playing kites on top of a machine . |
| | Two men sitting in a green room next to each other . |
| | There is a skate board on his beach with a tennis sky . |
| SeqGAN | A group of people on the beach that are uncooked . |
| | A man is sitting in a chocolate to he held on the beach . |
| | Two men are on the kite in the distance . |
| | A man is brushing his teeth on top of water . |
| | A group of zebra is seen on a video game . |
| | A group of guys standing at the beach during a frisbee . |
| | Two device with chairs and two ready for a women stands to it . |
| | The computer is playing a very steel skate games space . |
| | A man is using her pass . |
| | A group of men standing in a sky next to a dark building . |
| LeakGAN | A television brushing in a stair mat at the table . |
| | beautiful baseball player holding a bat is traveling in the ocean and water . |
| | Two young men standing near a phone at five coffee next to sidewalk . |
| | A blue dog sitting on top of a kite at a living room . |
| | A group of people standing with her kite in the beach . |
| | A big train with a checkered cake , keyboard and mouse . |
| | A person wearing a hat holding a wii boat holding two chairs . |
| | A man sitting next to a window . |
| | An old man holding a tennis ball while upside his racket in the sky by the sky . |
| | here is a boat on a living room on a table with a tennis ball . |

Table 3:  Samples taken from models on COCO Image Captions

| Sources | Example |
|---|---|
| OURS $\pi \rightarrow 0$ | in fact , this is in the past , at least in the european union , given the public nuclear program as a result of the macroeconomic ¡unk¿ . |
| | in the world , he lost an annual funds rate is even less than the government ' s single bank planning . |
| | the main factors view : preserving the us and central banks will be ¡unk¿ to the current global financial crisis , and is no longer . |
| | at this time , the us and the eurozone ' s ¡unk¿ would reduce the majority of policies that is still driven by both sides for ¡unk¿ . |
| | the democratic party will be able to ¡unk¿ the ¡unk¿ of the american and political and cultural opportunity to fix the security council . |
| | there has become a time for iran , and damaging only in the region where the us ¡unk¿ us economy would be competitive . |
| | after all , this is the ¡unk¿ of the lessons about the ¡unk¿ , would become the region seeking to challenge to the national culture . |
| | however , the budget deficits is not a true economic crisis in a growing global scale so much less than they have benefited from emerging economies . |
| | but they are in the crisis in other words , and it will have ¡unk¿ the emergence of a ¡unk¿ of the eu and north america . |
| | in the wake of the transformation of the media , there is a question who would be ¡unk¿ in the past 30 years . |
| OURS $\pi \rightarrow 1$ | in the rest of the election , however the us will not at the difficulty of climate change . |
| | more important should be the ¡unk¿ of europe and used in modern history . |
| | and all the chinese central banks was shown to ¡unk¿ the european union and the global financial crisis . |
| | with the global crisis , the us administration leaders must act fast in the ¡unk¿ of the global financial system . |
| | how many of the europeans could bring it to be in the common way to dominate the financial crisis . |
| | in fact , it is not at the same time , there is ¡unk¿ - by the west , and the global financial crisis . |
| | however , it is particularly important to explain the chinese government . |
| | this would be able to ¡unk¿ its debts , it is supposed to ¡unk¿ the international economy and ¡unk¿ violence . |
| | the costs of the ¡unk¿ will be increasing over the financial crisis . |
| | the world politics is likely to accept the expense of others in the crisis . |

Table 4: Samples taken from our proposed model on EMNLP 2017 News

| Sources | Example |
|---------|---------|
| MLE | they pushing for long as 18 % of chinese without the currency support of the new middle east ' s strengthening their imports . 

 the government ' s relationship with american power was ¡unk¿ to become a crucial tool to create a serious threat from israel and the ¡unk¿ of the way . 

 for example , however , the rest of the world is  we are the ¡unk¿ of the ¡unk¿ that markets must be able to success . 

 in the fourth quarter of 2011 , the largest of the world  s periphery economies created the world peace in the ¡unk¿ of 2007 . 

 the us was unable to its willingness to tackle the financial crisis , the new government must be addressed by the start to ¡unk¿ and the future . 

 indeed , the integrity of the world is not about whether the eu has to be ¡unk¿ by ¡unk¿ , as well as the iranian constitution , as well as their problem . 

 the us policy is , that would have not seen the dollar as the us to ¡unk¿ asia and the world economy . 

 moreover , the eu  s inflation is more likely to take more improved than about the us , and the benefits of economic recovery . 

 a key question is that a ¡unk¿ should be on a new framework for a ¡unk¿ , from the american government . 

 this is that the russian economy is just one-third of the value of $ 1 billion annually  a source of global trade . |
| SeqGAN | like the ¡unk¿ of the center for the most of the russian crisis . 

 the ¡unk¿ of the uk is right to ensure that the eu will ¡unk¿ . 

 it said that it says ¡unk¿ and could not take a significant impact on oil . 

 in asia , political has been wrong to the world bank , like what is the most important time . 

 in the recent crisis , is so that something else about most of the parties . 

 what is important for observers that high and how he could ¡unk¿ to support the education system . 

 we could develop a relatively little chance to fundamental classes , it is likely to ¡unk¿ its response to the yen . 

 on the asian financial system , like all of the west bank , it is what could not ¡unk¿ to put up its foreign ¡unk¿ . 

 in short , an unprecedented increase in the ecb would allow ¡unk¿ opportunity to rise to ¡unk¿ for the euro . 

 to manage the ¡unk¿ of the economic crisis , and the administration should increase their lives . |
| LeakGAN | but , gradually , the candidates must have to the excessive ¡unk¿ and that the old conflict can be made for the necessary housing . 

 both of the tax increase are on an opportunity to come to wonder that iraq is not little scenario . 

 ¡unk¿ to recognize the need for greater responsibility for ¡unk¿ concerning its foreign program . 

 finally , ¡unk¿ much of there is no more risky to achieve the goal . 

 it is likely to be reduced to make domestic risk to stimulate inflation in the national cooperation . 

 the relationship between a financial system is associated to raise income inequality . 

 but any recent problem is that made such events as much by ¡unk¿ up between global investment and strategic costs . 

 for radical institutions to keep the access to minimize the biggest impact of an independent world . 

 the ¡unk¿ process of capital requirements must be given that , or ¡unk¿ , and from the decision to guide global security . 

 it is nothing that hope that , lack of trust , such as a freedom of law . |

Table 5: Samples taken from baseline models on EMNLP 2017 News

