# OpenReview forum: "A Quality-Diversity Controllable GAN for Text Generation"
_ICLR.cc/2020/Conference — Reject_

### Official Review · AnonReviewer2 · 2019-10-17
**Official Blind Review #2**

**Rating:** 1

**Review:**

This paper proposed to use KL and reversed KL as its new objective function for text generation GAN training.
However, this paper missed a lot important references. Basically, the authors only compare results with seqGAN and leakGAN. MaliGAN (https://arxiv.org/pdf/1702.07983.pdf), TextGAN (https://arxiv.org/pdf/1706.03850.pdf), etc.
Also, KL + reversed KL training method for GAN framework is first proposed in Symmetric VAE (https://arxiv.org/abs/1709.01846), and the Proposition 2 basically are the same as the Symmetric VAE paper.

Therefore, I think this work is lack of novelty, and still need more time to work on.

**Experience Assessment:**

I have published one or two papers in this area.

**Review Assessment: Checking Correctness Of Derivations And Theory:**

I carefully checked the derivations and theory.

**Review Assessment: Checking Correctness Of Experiments:**

I carefully checked the experiments.

**Review Assessment: Thoroughness In Paper Reading:**

I read the paper thoroughly.

---

> ### Author Response · Authors · 2019-11-15
> **Response to Reviewer #2**
>
> Thank you for your comments. We'll add TextGAN and Symmetric VAE in our references. As for the MaliGAN, as far as we know, its proof for Theorem 3.1 is not correct, that's one reason that it couldn't be officially published.
>
> Regarding to the work of symmetric VAE, as shown in Section 4.2 of the symmetric VAE paper,  it is equivalent to GAN, where there are two sets of neural network parameters, one for generator, one for discriminator. But in N3DGAN proposed by Li et al., there is only one set of parameters for the generator and there is no neural network used for the discriminator, that's the major distinction with many GANs.
>
> The quality-diversity controllable GAN in this work is a generalization of N3DGAN proposed Li et al., 2019 and has connections to the forward KL divergence or the reverse KL divergence in terms of empirical distribution and model's distribution when pi is approaching 0 or 1, where the reverse KL divergence is not well defined since it has a term of log (p_g/0). KL and reversed KL have different meanings in symmetric VAE.

---

### Official Review · AnonReviewer1 · 2019-10-19
**Official Blind Review #1**

**Rating:** 1

**Review:**

This paper introduces a GAN-based text generation approach, where the authors propose to directly optimize a weighted version of JSD replacing p_data with its empirical distribution. I find the theoretical analysis of the approach confusing, thus would like to get clarification from the authors. The experiments largely rely on automatic evaluation, which is known to be unreliable for text generation. I'd like to see human evaluation of the generated sentences, and at least some example outputs should be shown (even if it's cherry-picked). Given that both the theory and the empirical results are not solid in the current version, I intend to reject the submission.

=== Theoretical analysis ===
1. Main question: based on Equation 2, the optimal solution p_G^* is the empirical data distribution. It's unclear if p_G^* goes to the real data distribution when N goes to infinity.
2. Given the definition of the generalized JSD in Equation y, when \pi = 0 and \pi = 1, JSD_\pi is both 0. How does it control the balance between forward and reverse KL? I'm also wondering what's the connection between Proposition 2 and the interpolation between forward and reverse KL (which is implied in the text).
3. In the proof of Proposition 2, what is H? Also in Equation 8, second line, how does the second term disappear? Would be good to have complete proof in the appendix.

=== Empirical results ===
1. The description of NLL_test and NLL_oracle is very brief. Could you specify what are the language model and data used in each case?
2. In Table 2, all numbers are pretty close, are they significantly different? It would be really helpful to show some qualitative results as well.
3. Is there evidence that \pi is controlling the tradeoff between quality and diversity? From the experiments it's mainly controlled by the temperature.

**Experience Assessment:**

I have published one or two papers in this area.

**Review Assessment: Checking Correctness Of Derivations And Theory:**

I carefully checked the derivations and theory.

**Review Assessment: Checking Correctness Of Experiments:**

I assessed the sensibility of the experiments.

**Review Assessment: Thoroughness In Paper Reading:**

I read the paper at least twice and used my best judgement in assessing the paper.

---

> ### Author Response · Authors · 2019-11-15
> **Response to Reviewer #1**
>
> Thanks for your detailed review.
> === Theoretical analysis ===
> 1、D_G^* has a term of empirical distribution, whether that term becomes real distribution when N goes to infinity is an open question, it will be answered in our future work.
> 2、Although generalized JSD is 0 when \pi = 0 and \pi = 1, generalized JSD/\pi and generalized JSD/(1-\pi) tend to be forward KL divergence and reverse KL divergence respectively when pi tends to 0 and 1.
> Generalized JSD is not simpy the interpolation between forward and reverse KL divergences.
> Reverse KL divergence is not well defined and we can't optimize it directly, but we can approximately optimize the reverse KL divergence by letting pi tend to 1. Models trained via forward KL divergence have a tendency to overgeneralise and generate unplausible samples which means diversity and models trained via reverse KL divergence will try to avoid any behaviour that is unlikely under data distribution which means quality.
> 3、H is the positive definite Hessian. We will give more detailed proof in the final version.
> === Empirical results ===
> 1、For NLL_oracle, we take as input the sentences generated by the language model we trained to the oracle LSTM model. For NLL_test, we take as input the sentences generated by the oracle LSTM model to the language model we trained.
> 2、we give more detailed qualitative analysis of Table 1 and Table 2 in the final version.
> 3、As can be seen from figure 1 and figure 3, given the same list of temperature, the language models trained with different \pi have notably distinct performances on quality and diversity metrics.We also add example outputs for both COCO and EMNLP 2017 News tasks in the final version and we can see that our proposed model performs better than SeqGAN and LeakGAN.

---

### Official Review · AnonReviewer3 · 2019-10-22
**Official Blind Review #3**

**Rating:** 3

**Review:**

This paper provides a method (loss function) for training GAN model for generation of discrete text token generation. The aim of this loss method to control the trade off between quality vs diversity while generating the text data.

For example,
if original sentence is "The company ’ s shares rose 1 . 5 percent to 1 . 81 percent , the highest since the end of the year ." and the output is "The company ’ s shares rose 1 . 5 percent to 1 . 81 percent , the highest since the end of the year ." then the quality of generation is high and diversity is low.

Pros:
1. The paper is very well written, with importance to the smaller details. It is a very good read for even people who are new to this problem. Especially, I appreciate the part where authors took efforts in writing why a few metrics are not used!
2. The motivation is good and also contributions are explicitly written. The details of the approach are provided clearly.
3. The experiments are provided in two different datasets and also the experiments support the two major claims in the paper.

Cons:
1. The primary concern with this submission is the novelty. The Proposition 1 of using forward-backward JSD based divergence has already been proposed in Li et al. (2019). Also, Li et al. (2019) proposes the entire contribution of this paper. The only difference is the introduction of \pi, which controls the percentage of labelled data to be considered between the generated data and original data. Basically, Li et al. (2019) is a specific case of this paper where \pi = 0.5. Thus, I would consider this paper as one additional experiment in Li et al. (2019) and not a whole paper as such.
2. Also, in the formulation in Eqn 2, the proposition 2 becomes a direct observation when \pi becomes 0 or \pi becomes 1. I would not call this as a proposition but a mere observation of Eqn 2.
3. Thus, taking away proposition 1 (already proposed in Li et al. (2019)) and proposition 2 (which is a mere observation) I do not find any novelty in this paper.
4. From an experiments perspective, Li et al. (2019) performed experiments in 4 datasets: Chinese Poems, MS COCO captions, Obama Speech, and EMNLP2017 WMT news. However, in this paper, results are shown in only two datasets - MS COCO captions and EMNLP2017 WMT news. Was it because that this paper was submitted in haste and/or the results in the other two datasets are not compelling enough to share?
5. The result analysis are poor - the authors have shown only the numbers in the tables, while the interpretation on these numbers and the discussion is left to reviewers discretion. Also, there are no generated examples that the authors are showing in either of the datasets. The authors should further discuss and analyze the results, show generated examples, and explain success and failure cases and the reasons behind them.

Overall, I find the novelty and the experimental analysis of the paper, very weak.

**Experience Assessment:**

I have published one or two papers in this area.

**Review Assessment: Checking Correctness Of Derivations And Theory:**

I carefully checked the derivations and theory.

**Review Assessment: Checking Correctness Of Experiments:**

I assessed the sensibility of the experiments.

**Review Assessment: Thoroughness In Paper Reading:**

I read the paper thoroughly.

---

> ### Author Response · Authors · 2019-11-15
> **Response to Reviewer #3**
>
> Thanks for your detailed review.
> 1、We introduce \pi and demonstrate that quality and diversity can be controlled by using different \pi.
> In addition, Li et al. (2019) don't use temperature sweep in the experiments, but only report performances under quality metric at temperature=1.
> 2、Thanks for pointing out a more concise way to prove proposition 2. Although the process is concise, we think that the conclusion is interesting, which shows that we can control the dependence on forward KL divergence and reverse KL divergence by controlling a single hyperparameter \pi, thereby achieving control over quality-diversity trade-off.
> 3、Please refer to the above responses to the first and second questions.
> 4、We chose COCO dataset and EMNLP2017 WMT News dataset since they have become common benchmarks for text generation.The related papers such as "Language GANs falling short", "Jointly measuring diversity and quality in text generation models", "Training language gans from scratch" and "Neural Text Generation: Past, Present and Beyond" all utilize both or one of these two datasets.
> 5、We add the generated sentences for both COCO and EMNLP 2017 News tasks and corresponding analysis in the final version, and we can see that our proposed model performs better than SeqGAN and LeakGAN.
> We also give more detailed qualitative analysis of Table 1 and Table 2 in the final version.

---

### Decision · Program_Chairs · 2019-12-19

**Decision:**

Reject

**Comment:**

This paper provides a method (loss function) for training GAN model for generation of discrete text token generation. The aim of this loss method to control the trade off between quality vs diversity while generating the text data.

The paper is generally well written, but the experimental section is not overly good: Interpretation of the results is missing; error bars are missing.